# Cobamide-mediated enzymatic reductive dehalogenation via long-range electron transfer

Cindy Kunze[1,*], Martin Bommer[2,*,†], Wilfred R. Hagen[3], Marie Uksa[1,†], Holger Dobbek[2], Torsten Schubert[1] & Gabriele Diekert[1]

The capacity of metal-containing porphyrinoids to mediate reductive dehalogenation is implemented in cobamide-containing reductive dehalogenases (RDases), which serve as terminal reductases in organohalide-respiring microbes. RDases allow for the exploitation of halogenated compounds as electron acceptors. Their reaction mechanism is under debate. Here we report on substrate–enzyme interactions in a tetrachloroethene RDase (PceA) that also converts aryl halides. The shape of PceA's highly apolar active site directs binding of bromophenols at some distance from the cobalt and with the hydroxyl substituent towards the metal. A close cobalt–substrate interaction is not observed by electron paramagnetic resonance spectroscopy. Nonetheless, a halogen substituent *para* to the hydroxyl group is reductively eliminated and the path of the leaving halide is traced in the structure. Based on these findings, an enzymatic mechanism relying on a long-range electron transfer is concluded, which is without parallel in vitamin $B_{12}$-dependent biochemistry and represents an effective mode of RDase catalysis.

[1] Department of Applied and Ecological Microbiology, Institute of Microbiology, Friedrich Schiller University, Philosophenweg 12, Jena D-07743, Germany. [2] Structural Biology/Biochemistry, Institute of Biology, Humboldt Universität zu Berlin, Philippstrasse 13, Berlin D-10115, Germany. [3] Department of Biotechnology, Faculty of Applied Sciences, Delft University of Technology, van der Maasweg 9, Delft 2629HZ, The Netherlands. * These authors contributed equally to this work. † Present address(es): Max-Delbrück-Centrum for Molecular Medicine, Robert-Roessle-Str. 10, Berlin D-13092, Germany (M.B.); Institute of Soil Science and Land Evaluation, University of Hohenheim, Emil-Wolff-Strasse 27, Stuttgart D-70593, Germany (M.U.). Correspondence and requests for materials should be addressed to T.S. (email: torsten.schubert@uni-jena.de) or to G.D. (email: gabriele.diekert@uni-jena.de).

Several anaerobic bacteria use organohalides as terminal electron acceptors in their respiratory metabolism. These often toxic, hazardous and usually highly persistent compounds, which originate from industrial, biotic or geochemical sources, are reductively dehalogenated by these microbes. This biological process mobilizes the halogens and counteracts the accumulation of organohalides in oxygen-depleted environments. Hence, organohalide respiration contributes significantly to the global halogen cycle. Reductive dehalogenase (RDase) enzymes are membrane-bound terminal reductases in organohalide respiration and harbour two Fe–S clusters and a cobamide cofactor (reviewed in ref. 1). The utilization of a cobamide cofactor makes RDases unique among terminal reductases. With almost 300 RDase genes identified so far, organohalide respiration is present in different bacterial phyla, including Chloroflexi, Firmicutes and Proteobacteria[2]. However, only a dozen of the corresponding gene products were biochemically characterized including the tetrachloroethene RDase (PceA) of the epsilonproteobacterium *Sulfurospirillum multivorans*[3]. PceA was described to mediate the reductive dehalogenation of chlorinated and brominated ethenes or propenes[4,5] (Supplementary Fig. 1). The cobamide cofactor of PceA was identified as norpseudo-B$_{12}$, a derivative of vitamin B$_{12}$, which is characterized by a unique nucleotide loop composition[6]. The crystal structure of PceA showed the norpseudo-B$_{12}$ non-covalently bound in its 'base-off' conformation deeply inside the protein[7]. The two [4Fe–4S] clusters of PceA connect the surface and cobamide cofactor at distances short enough to allow intramolecular electron transfer to the active site and potentially also from the proximal Fe–S cluster to the substrate. An identical arrangement of the metal cofactors was detected in the *ortho*-dibromophenol RDase (NpRdhA) of the marine alphaproteobacterium *Nitratireductor pacificus* pH-3B[8], a non-respiratory RDase with 28% amino acid sequence identity to PceA.

Apart from methyltransferases, adenosylcobalamin-dependent enzymes (for example, eliminases, mutases and ribonucleotide reductase) and *S*-adenosylmethionine radical enzymes[9–11], RDases form a distinct subfamily of cobamide-dependent enzymes together with the epoxyqueuosine reductase[12]. Cobamide-dependent methyltransferases heterolytically cleave the cobalt–carbon bond in methylcobalamin and transfer a methyl ion. Adenosylcobalamin-dependent enzymes generate a 5′-deoxyadenosyl radical via homolytic cleavage of the Co–C bond. The adenosyl radical then serves as reactive species during catalysis. Different from these extensively investigated modes of cobamide cofactor function, recently alternative mechanisms have been proposed for the cobamide-dependent *S*-adenosylmethionine radical enzymes catalysing either methylations or substrate rearrangements[13,14].

Little is known about the catalytic mechanism of RDases that harbour derivatives of hydroxocobalamin or aquocobalamin rather than adenosylcobalamin or methylcobalamin as cofactors. The super-reduced [Co$^I$]-state was proposed to initially attack the substrates[7,8]. However, different reaction mechanisms for cobamide-dependent reductive dehalogenation have been proposed (Fig. 1). The formation of a cobalt–carbon bond[15] after alkylation of the cobalt by a nucleophilic attack of [Co$^I$] on the carbon backbone of the organohalide, the formation of a cobalt–halogen bond after direct [Co$^I$] attack on the halogen substituent[8] or a long-range electron transfer from [Co$^I$] leading to substrate radical formation followed by the formation of a carbanion after elimination of the halogen substituent[5,7,16] were considered. As revealed by the structural analysis of PceA and NpRdhA, an alkylation of the cobalt during substrate conversion is unlikely due to spatial restraints caused by the amino acid arrangement at the active site[7,8]. For NpRdhA, spectroscopic analysis and substrate modelling pointed towards the formation

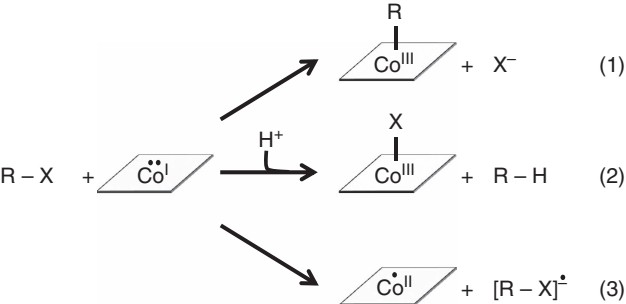

**Figure 1 | Proposed initial steps in the catalytic mechanism of RDases.** (1) Alkylation of the cobalt by a nucleophilic attack of [Co$^I$] on the carbon backbone of the organohalide. (2) Formation of a cobalt–halogen bond after [Co$^I$] attack directly at the halogen substituent followed by heterolytic cleavage of the carbon–halogen bond. (3) Long-range electron transfer from [Co$^I$] leading to substrate radical formation and finally to the formation of a carbanion after elimination of the halogen substituent (the elimination is not shown). [Co$^{I–III}$]: oxidation states of the cobalt ion in the cobamide cofactor; R: hydrocarbon backbone; X: halogen substituent.

of a cobalt–halogen bond during the conversion of brominated aromatic substrates[8]. The structure of PceA occupied by trichloroethene (TCE) showed cofactor distances of 5.8 Å to the cobalt, 10.8 Å to the proximal [4Fe–4S] cluster and 3.7 Å to a potential proton donor, a highly conserved tyrosine[7]. However, the small size of the ligand, two different orientations and the lack of a defined binding site did not allow for an unambiguous deduction of the course of the reaction. A reorientation of the substrate in the reduced enzyme and a direct interaction of the substrate with the cobalt would have been feasible, even though this was not observed in the crystal structure. So far, there is no direct evidence for a long-range electron transfer in RDases, but there are indications for an undirected electron transfer mechanism in PceA during reductive dehalogenation of tribromoethene (TBE) to all isomers of dibromoethene (DBE)[5] and of *trans*-1,3-dichloropropene to a mixture of *cis*-1-, *trans*-1- and 3-chloropropene[4] (Supplementary Fig. 1). In addition, adduct formation with radical traps or chloropropenyl radicals was observed during the conversion of chloropropenes and interpreted as indicative for the generation of substrate radicals[16].

In this study, the binding and conversion of brominated and chlorinated phenols by PceA of *S. multivorans* is investigated in detail. These organohalides are chosen for three reasons: (i) The active site cavity of PceA allows for the binding of larger halogenated phenols, which might overcome the ambiguity in substrate orientation. (ii) Specific halogenated phenols could be selected, whose substitution pattern presumably would allow probing of potential substrate–cobalt interactions. (iii) Halogenated phenols are applied for a direct comparison with RDases specialized in converting aromatic organohalides. Here we show that PceA is also able to convert brominated phenols. Structural analysis of PceA–substrate complexes displays the hydroxyl substituent positioned above the cobalt at a distance of 4.7 Å. From the absence of a direct cobalt–substrate coupling during substrate turnover, which is verified by electron paramagnetic resonance (EPR) spectroscopy, an attack via a long-range electron transfer is concluded as initial step. The reductive dehalogenation at the active site initiated by the dissociative electron transfer is visualized in the PceA crystal.

## Results

**Dehalogenation of bromophenols (BPs) by PceA**. Structural analysis showed the active site of PceA located at the centre of the enzyme[7]. The narrow, triangular-shaped substrate-binding

**Table 1 | Conversion of halogenated substrates by purified Strep-tagged PceA isolated from S. multivorans GD21 and the effect of ammonium on the reaction.**

| Substrate | Product | Without $NH_4^+$ | | | With $NH_4^+$ | | Fold increase in activity with $NH_4^+$ |
| --- | --- | --- | --- | --- | --- | --- | --- |
| | | $V_{max}$ (nkat mg$^{-1}$ PceA) | $k_{cat}$ (s$^{-1}$) | $k_{cat}/K_m$ (mM$^{-1}$ s$^{-1}$) | $V_{max}$ (nkat mg$^{-1}$ PceA) | $k_{cat}$ (s$^{-1}$) | |
| PCE | TCE | 1,008 ± 24 | 54 | 270 | 3,363 ± 47 | 180 | 3.4 |
| TCE | cis-1,2-DCE | 890 ± 4 | 48 | 200 | 3,613 ± 278 | 193 | 4.1 |
| 2-BP | phenol | 38,108 ± 1,660 | 2,035 | ND | 39,934 ± 14 | 2,132 | 1 |
| 3-BP | phenol | 9,204 ± 256 | 492 | ND | 8,475 ± 1,534 | 453 | 0.9 |
| 4-BP | phenol | 15,143 ± 332 | 809 | 8,172 | 15,332 ± 27 | 819 | 1 |
| 2,4-DBP | 4-BP | 4,653 ± 229 | 252 | 2,655 | 5,975 ± 50 | 319 | 1.2 |
| 2,5-DBP | 3-BP, 2-BP | 40,704 ± 12,183 | 2,173 | ND | 40,420 ± 5,256 | 1,898 | 0.9 |
| 2,6-DBP | 2-BP | 14,176 ± 0 | 757 | ND | 15,871 ± 2,059 | 848 | 1.1 |
| 3,5-DBP | 3-BP | 152 ± 0.3 | 8 | ND | 167 ± 30 | 9 | 1.1 |
| 2,4,6-TBP | 2,4-DBP | 107 ± 46 | 8 | 53 | 102 ± 21 | 5 | 0.65 |
| 2,3-DCP | 3-CP | 43 ± 0 | 2 | ND | 35 ± 0 | 2 | 0.8 |
| 2,5-DCP | 3-CP | 6 ± 1 | 0.3 | ND | 5 ± 0.5 | 0.3 | 1 |

ND, Km was not determined.
$k_{cat}$ of dibromophenols comprised the formation of the corresponding bromophenol as well as further reduction to phenol, whereby all turnovers were included in its calculation. S.d. is given. No dehalogenating activity was measured with 2-, 3- or 4-CP, 2,4-DCP, 2,6-DCP, 3,4-DCP, 3,5-DCP, 2,3,4-TCP, 2,4,5-TCP, 2,4,6-TCP and 3,4,5-TCP, with the larger halogenated aromatics 3-chlorobenzoate, 3-chloro-4-hydroxyphenyl-acetate or with cis-1,2-dichloroethene (cis-1,2-DCE). The variety of bromophenols tested here was limited to commercially available compounds. The detection limit of phenolic compounds was 5 µM. $K_m$ for PCE is 0.2 mM and for TCE 0.24 mM[3].

pocket with its base above the corrin ring and its maximal height above the cobalt ion, both dimensions about 10 Å in length, easily allows access for chlorinated and brominated ethenes and propenes, which are the known substrates of PceA[3–5]. In order to investigate structural restrictions in the active site that are responsible for substrate selectivity and substrate positioning, the substrate range of PceA was revisited and broadened towards bulkier electron acceptors. Enzyme activity measurements revealed that besides aliphatic hydrocarbons brominated and chlorinated phenols are also converted by the enzyme (Table 1). The halogenated phenols are expected to be readily accessible for the reduction by the super-reduced $[Co^I]$ of PceA (midpoint potential of the $[Co^{II}]/[Co^I]$ couple: $-380$ mV at pH 7.5 (ref. 6)), because of their positive redox potentials (E°′ $= 300$–$500$ mV)[17]. All tested brominated phenols were completely reduced to phenol. BPs were converted with turnover numbers up to $\sim 2,000$ s$^{-1}$, thus 40-fold higher compared to the $k_{cat}$ of 54 s$^{-1}$ for tetrachloroethene (perchloroethylene (PCE)). While most of the brominated phenols were selectively dehalogenated, 2,5-dibromophenol (DBP) was debrominated to 3- and 2-BP. Conversion rates about 20–30 times lower than for PCE were measured for 3,5-DBP and 2,4,6-tribromophenol (TBP). In general, PceA preferentially removed the bromine substituent at the ortho-position followed by the halogen substituent at either the meta- or para-position. With the exception of the 3,5-dichlorophenol RDase[18] of Desulfitobacterium hafniense PCP-1, all biochemically characterized RDases favour the removal of the ortho-substituent of chlorinated phenols and preferentially convert polyhalogenated phenols[19–22]. In case of PceA, the conversion rate increased with a decrease in the number of halogen substituents, as shown for 2,4,6-TBP to 2,4-DBP and 4-BP, while the $K_m$ values for all three substances were similar. The apparent $K_m$ for 4-BP was 99 µM, for 2,4-DBP 95 µM and for 2,4,6-TBP 158 µM. Substrate concentrations $> 600$ µM for 2,4-DBP to 1,000 µM for 2,4,6-TBP inhibited PceA. RDases have not been previously described to use both alkyl and aryl halides. For the chlorophenol RDases of Desulfitobacterium dehalogenans and Desulfitobacterium hafniense DCB-2, a dehalogenation of chlorinated ethenes was detected but at low rates[19,20]. PceA being an RDase that converts both types of substrates at similarly high rates, it allows for mechanistic studies on a single RDase reductively dehalogenating

alkyl and aryl halides. The efficient dehalogenation of brominated phenols by the PceA enzyme sheds a new light on its role in nature that has been defined so far as an effective catalyst for the dehalogenation of alkyl halides such as PCE and TCE, both substrates of mainly anthropogenic origin. In contrast to brominated phenols, PceA did not dehalogenate most of their chlorinated analogues. A similar preference for brominated substrates rather than their chlorinated counterparts has been reported for NpRdhA[8]. PceA dechlorinated only 2,3- and 2,5-dichlorophenol (DCP) with a $k_{cat}$ of 1.9 s$^{-1}$ and 0.3 s$^{-1}$, respectively. Both substrates were exclusively dehalogenated at the ortho-position. The formation of phenol was not observed with either substrate.

PceA dehalogenated 4-iodophenol (4-IP) three times faster than 4-BP, while 4-chlorophenol (4-CP) was not converted. The dehalogenating activity increased with decreasing electronegativity and decreasing partial negative charge from the chlorine to the iodine substituent. Partial charge models and Gibbs free energy calculations for chlorinated and brominated organohalides provide a rationale for these observations[23,24]. Previous studies on PceA of S. multivorans revealed that the presence of ammonium ions stimulates the conversion of halogenated ethenes[3,5]. To test the effect of ammonium ions on the reduction of halogenated phenols, 4 mM $(NH_4)_2SO_4$ was added to the assay. Conversion of PCE was stimulated 3.3-fold, but no effect on the turnover of halogenated phenols was observed (Table 1). This difference was also described earlier for chlorinated propenes compared to chloroethenes[16]. However, the positive effect of ammonium ions on PCE conversion remains inexplicable. The different conversion rates depending on the redox potential of the artificial electron donor described earlier by Miller et al.[25] for PCE were confirmed here for 4-BP (Supplementary Table 1).

The conversion of substrates was strictly dependent on the intact enzyme, involving the super-reduced $[Co^I]$-state of the cobamide cofactor. No abiotic conversion of any substrates mediated by protein-free cobamides was detected, even when heat-inactivated PceA was applied in a 120- to 160-fold concentration compared to that of native PceA. However, the involvement of the enzyme-bound cobamide cofactor in the catalysis was corroborated by the complete inhibition of the PceA-mediated 4-BP dehalogenation by propyl iodide in the dark. Propyl iodide is an inhibitor that binds irreversibly to the

[Co$^{I}$]-state of cobamides in the absence of light[26,27]. Subsequent exposure to light reversed the inhibition of 4-BP conversion. The inhibition of the dehalogenating activity of cobamides by propyl iodide has been attributed to an alkylation of the [Co$^{I}$]-state, which implies the formation of a cobalt–carbon bond for the propyl iodide probe. However, this is not the case for phenolic substrates as will be shown below.

**Binding of halogenated phenols in the active site cavity.** The conversion of halogenated phenols by PceA raised the question of their positioning in the active site. With respect to their dimensions, monoaromatic substrates with several substituents should fit into the active site of PceA. However, due to the arrangement of the amino acid side chains, aryl halides are expected to be limited in their orientation. To test for the validity of this hypothesis, PceA crystals harbouring the five coordinated [Co$^{II}$]-state of the cobamide cofactor were soaked with halogenated phenols and the 3D structure of the enzyme–substrate complex was determined and analysed. Several monoaromatic halogenated compounds up to the size of 2,4,6-TBP were visualized in the active site pocket. Restricted by the protein environment, the phenol ring of 2,4,6-TBP is oriented at an angle of approximately 40° away from the surface normal vector of the corrin ring and enclosed by Trp96, Tyr382, Trp56, Trp376, Tyr102 and Tyr246 (Fig. 2a). The substrate enters the active site through an opening between Phe38, Trp376, Tyr102 and Asn272 at the bottom of the substrate channel. The binding pocket is thus lined with hydrophobic side chains, while the substrate is shielded from the polar protein backbone. At the far end of the hydrophobic-binding pocket, a gap of the size of a single halide atom between residues Tyr102, Trp56 and Tyr382 (arrows in Fig. 2a) allows access to a polar upper cavity containing the hydroxyl groups of Tyr102 and Tyr382, as well as Glu92, Lys64 and three water molecules. The hydroxyl group of 2,4,6-TBP is located 4.7 Å away from the cobalt and 2.4 Å from the Tyr246-OH. The bromine substituent at C2 is fully enclosed, while bromine at C6 points towards the substrate channel and the bromine at C4 towards the upper cavity, where it is restrained in its position by Tyr102 and Tyr382. The hydrogen at C3 points at the Tyr382 phenyl group. The two rings interact in an edge-to-face geometry, precluding a bulky halogen substituent at this place. In addition, C$_{beta}$ of Tyr102 is only 3.3 Å away from C5 of the aromatic substrate ring, leaving little space for an additional halogen substituent. Hence, binding of the planar, triangular 2,4,6-TBP to PceA with a bromine substituent towards Co is likely to be impossible (see Fig. 2 and Supplementary Fig. 2a,b) and would place the other bromine substituents in conflict with the aromatic ring of Tyr382 or into the substrate channel between Tyr102 and Phe38, a position too narrow to accommodate a bromine. While the latter may be resolved by not-yet-observed plasticity within the binding site, the short distance between the conserved Tyr246 and the substrate hydroxyl group (2.4 Å) in the observed position should be noted. This is likely to prevent the binding of the bulkier bromine substituent at the current hydroxyl position or closer in not only 2,4,6-TBP but also many other brominated phenols tested in this study. The low $K_m$ value for 2,4,6-TBP supports the reliability of the analysed enzyme–substrate complex, since it indicates a high affinity for the substrate without any steric hindrance in substituent positioning.

Based on the position of the aromatic ring of 2,4,6-TBP, a tentative model for the probability of a halogen substitution at the different C-atoms in monoaromatic organohalides was drafted as working hypothesis (Fig. 2b). This model suggests that the substitution pattern governs the substrate orientation in the binding pocket and might thereby influence the reactivity. When

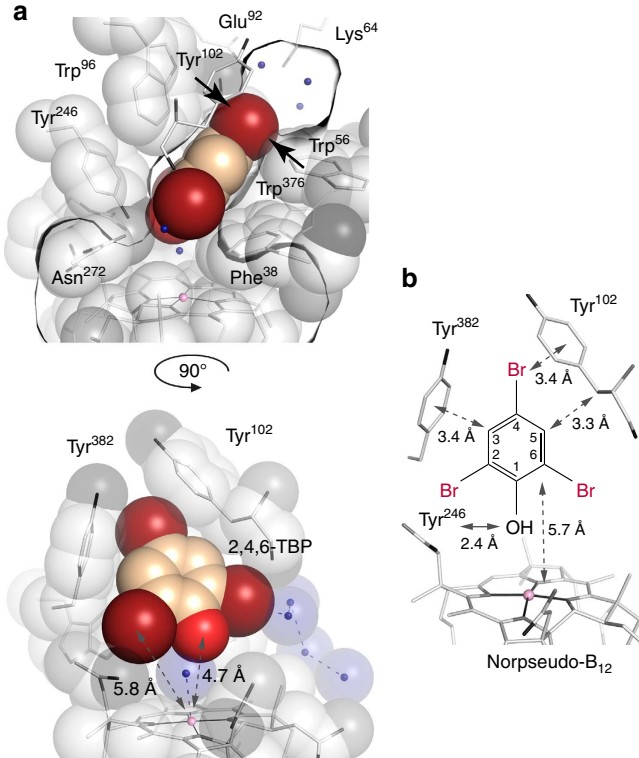

**Figure 2 | The PceA substrate-binding pocket. (a)** Positioning of 2,4,6-TBP in the active site as revealed by structural analysis. Shown is a cross-section through the binding pocket viewed from the substrate channel (Tyr102) (upper figure) or viewed from Tyr246 with the substrate channel on the right (lower figure). The Co-water/hydroxyl ligand and further water molecules are shown as blue spheres. **(b)** Distances between 2,4,6-TBP and Tyr102, Tyr246, Tyr382 and cobalt. A single hydrogen bond may be formed to the phenolic hydroxyl of Tyr246. Probability of a halogen substituent at different C-atoms provides a tentative model for the binding of other monoaromatic substrates. Besides the allowed substitution positions shown, the Tyr102-oriented position is possibly allowed after a small rotation of the substrate. A substituent at the Tyr382-oriented position is not possible due to bulky obstruction.

crystals were treated with 2 mM Ti(III) citrate or up to 5 mM Eu(II)EDTA/DTPA and 0.2 mM methyl viologen, the electron density for the upper ligand of the cobamide cofactor disappeared, which was attributed to the presence of the super-reduced [Co$^{I}$]-state of the cobalt ion. Since there were no further conformational changes visible upon reduction of the substrate-free enzyme, other than lacking the upper ligand, it is likely that the substrates will be positioned the same way in the [Co$^{I}$]-enzyme. Unfortunately, the destabilizing effect of reducing agents onto PceA crystals occupied with substrates did not allow for an analysis of enzyme–substrate complexes under these conditions. Furthermore, when incubated with substrate and reducing agent, PceA crystals apparently reverted to the [Co$^{II}$] state as judged by their colour and the presence of the upper Co-ligand (Fig. 3a). In order to confirm the functionality of PceA in the crystals, activity assays were performed with crystallized PceA under the same buffer conditions used for crystal generation and storage. Reductive dehalogenation of 4-IP, 4-BP and TCE was observed upon reduction of the crystals with methyl viologen and Ti(III) citrate. Conversion of other substrates was not tested with crystals.

To verify our hypothesis on the binding mode of phenolic substrates presented in Fig. 2b and to understand the sequence and mechanism of organohalide reduction, PceA crystals in the

[Co$^{II}$]-state were soaked with 2,4-DBP and 4-BP, the dehalogenation products of 2,4,6-TBP (Fig. 3a and Supplementary Fig. 3). Interestingly, 2,4-DBP and 4-BP, which were expected to freely rotate around the vertical and horizontal axis, occupied the same position as 2,4,6-TBP. While the substrate hydroxyl groups were nearest to the cobalt at a distance of 4.7 Å, a ring formed by Arg305, Trp376, Asn272, Phe38 and the carboxamide side chains of the cobamide cofactor restricts access of the substrate to the metal ion. The substrate hydroxyl group is close to Tyr246-OH (2.4 Å), which additionally limits its approach to Co, thus disfavouring a coupling of the cobalt with the hydroxyl group of the substrate. All in all, the binding mode of halogenated phenols confirmed the previous assumption that an alkylation of the cobalt during the course of the reaction is unlikely. Moreover, the orientations of 2,4,6-TBP, 2,4-DBP and 4-BP contradict the formation of a cobalt–halogen bond and rather indicate a long-range electron transfer during catalysis. According to the prediction of allowed halogen positions (Fig. 2b), *meta*-halogenated phenols might not fit into the active site with the hydroxyl group pointing towards the cobalt. In 3-BP soaked crystals, the hydroxyl group is turned by one position towards the substrate channel/Tyr102 and the bromine substituent is oriented towards the upper cavity (Fig. 3b). None of the substituents are orientated towards the cobalt ion, precluding a direct attack.

In the first step of the reaction, the electrons could be externally transferred to the bromine substituent or to the aromatic ring. For 2,4,6-TBP, the bromine substituents adjacent to the hydroxyl group seem to be preferentially removed and could accept electrons from the aromatic ring as well as from the cobalt ion itself, considering their distance of approximately 5.8 and 6.6 Å from the cobalt. However, the *para*-position of 4-BP and the *meta*-position of 3-BP are placed distantly away, towards the upper cavity of the substrate-binding pocket, and yet reductive dehalogenation was not impaired. In these cases, electrons from [Co$^I$] have to be transferred via the aromatic ring to the respective substituent. Crystals were also soaked with 2,4,6-trichlorophenol and 3-CP (Supplementary Fig. 4a,b). Both phenols were positioned in the same orientation as their brominated analogues in the substrate-binding pocket but were not dehalogenated. Activity measurements using 4-CP, 4-BP and 4-IP showed an enhanced reduction rate with decreasing electronegativity of the halogen substituent from chloride to iodide. The position and orientation of the analogues 4-CP, 4-BP and 4-IP in the active site of substrate–enzyme complexes is thereby identical, independent of the type of halogen (Fig. 3c). It has to be concluded that exclusively the differences in the energy of the various carbon–halogen bonds determine the displacement or continuance of the substituent.

It should be noted that the hydrophobic substrate-binding pocket of PceA did not form hydrogen bonds with its substrates. Hence, the position of the substrate and its halide substituents is not strictly fixed. A subset of substrates or substrate analogues was identified, which did not match with the orientation of 2,4,6-TBP in the active site pocket (Supplementary Fig. 4c).

Though the dimensions of the active site allow for the binding of monoaromatic organohalides, access to the pocket is restricted by a 5.5 × 3 Å 'letter box' entry of the substrate channel, which is embedded within a hydrophobic groove on the protein surface and lined by the side chains of Thr39, Phe44, Phe57, Leu186 and Glu189 (ref. 7). These side chains appear to be immobile and were found at the same position in crystal structures of oxidized, substrate-bound and reduced PceA. 4-IP and 2,6-DBP bind to the groove in multiple positions, as shown at Leu186 (Supplementary Fig. 2b,c). The position of all letter box side chains is severely restrained and Leu186 is the only side chain, which from observation of possible side-chain rotamers, may move

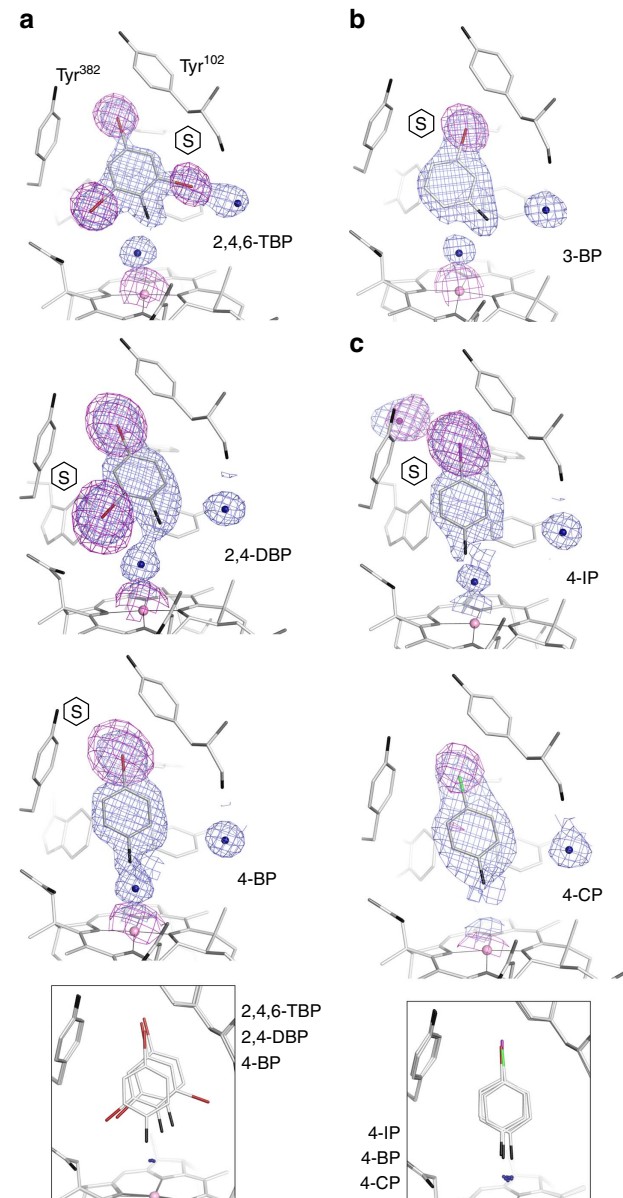

**Figure 3 | Substrates and analogues bound to the PceA substrate-binding pocket.** (**a**) Positioning of 2,4,6-TBP, 2,4-DBP and 4-BP. An overlay of the three substrates is depicted in the inset. Stereo representations are shown in Supplementary Fig. 3. (**b**) Positioning of the *meta*-halogenated 3-BP. (**c**) Influence of halogen type on positioning of *para*-halogenated phenols. In the direction shown, the binding site is flanked by Tyr382, Tyr102 and the corrin ring. For all substrates, 1 sigma $2F_o - F_c$ electron density (blue) for substrate, Co-ligand water/hydroxyl and the first water molecule in the substrate channel are shown. Anomalous difference density (indicative of a heavy atom) is shown around the substrate for bromine (5 sigma map, red stick), chlorine (3.5 sigma, green stick) or iodine (5 sigma, violet stick). The hydroxyl group is shown in black. (S) Indicates the leaving group of a substrate. Note that products were bound to PceA at high concentrations and represent substrate rather than product, except for a signal modelled as iodide ion which may have dissociated from 4-IP in the 1.9 Å wavelength X-ray beam. 2,4-DBP was incubated in a buffer containing 200 mM Cl$^-$; other incubations were performed in chloride-free buffer. 2,4-DBP and 4-BP were incubated in the presence of 2 mM Ti(III) citrate and 0.2 mM methyl viologen. No cobalt β-ligand was modelled for 4-CP because of the lower resolution (2.3 Å versus 1.6–1.9 Å for other structures).

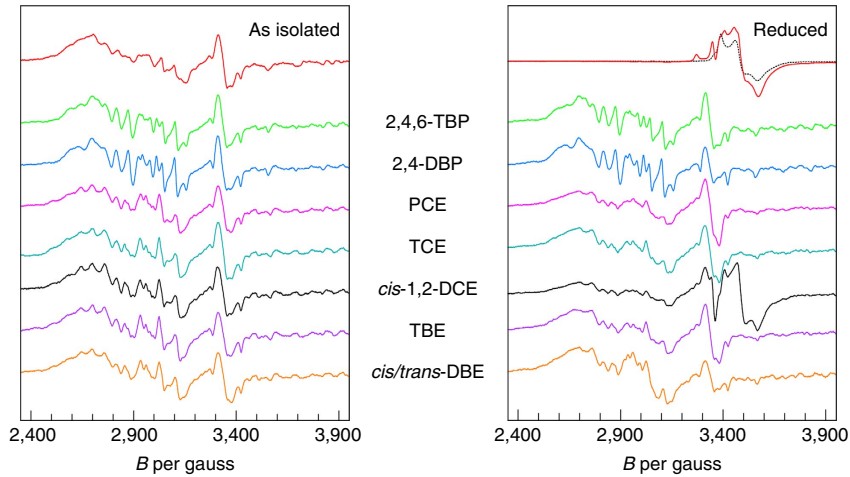

**Figure 4 | Stack plot of substrate-induced and turnover-induced changes in the [Co$^{II}$]-EPR of PceA.** The red trace in the left-hand panel is the spectrum of the as-isolated enzyme. Subsequent traces are for incubation of the as-isolated PceA with 2,4,6-TBP (green trace), 2,4-DBP (blue trace), PCE (magenta trace), TCE (cyan trace), *cis*-1,2-DCE (black trace), TBE (violet trace) and *cis/trans*-1,2-dibromoethene (*cis/trans*-DBE, gold trace). In the right-hand panel, the red trace is the spectrum of reduced PceA. Reduction of PceA (3 mg ml$^{-1}$) with 1 mM Ti(III) citrate and 20 μM methyl viologen for about 5 min led to a [4Fe–4S]$^{1+}$ spectrum accompanied by the spectra of Ti(III) citrate and a methyl viologen radical. The lack of a [Co$^{II}$]-signal indicated the reduction of the cobalt to the EPR silent [Co$^{I}$]-state. The black dotted line is the spectrum of Ti(III) citrate alone. The subsequent traces are the equivalents of the traces in the left-hand panel after turnover of substrates induced by reduction of PceA for about 5 min. Experimental EPR conditions were: microwave frequency, 9,338 MHz; microwave power, 12.7 mW; modulation frequency, 100 kHz; modulation amplitude, 8 Gauss; temperature, 22 K.

away from the opening. While such open conformation was not observed, Leu186 was poorly defined when crystals were incubated with 3-BP and a peak in the anomalous difference map suggests a partial occupation by a 3-BP bromine substituent (Supplementary Fig. 2d–f). Leu186 may thus shift to allow access of larger compounds such as 2,4,6-TBP to the active site.

**EPR spectroscopic analysis of PceA–substrate interactions.** To confirm the hypothesis of a long-range electron transfer mechanism during the reductive dehalogenation of halogenated phenols, enzyme–substrate complexes were studied by EPR spectroscopy. As-isolated PceA in solution showed the characteristic low-spin $S = 1/2$ spectrum of the [Co$^{II}$]-state of the cobamide cofactor (Fig. 4, left panel). The addition of 2,4,6-TBP and 2,4-DBP did not change the [Co$^{II}$]-signal significantly (Fig. 4, green and blue tranches). An increase in resolution was observed that was accompanied by minor changes in the effective spin Hamiltonian parameters, which are estimated to be <0.1% in the *g*-value and a few gauss in the hyperfine splittings (see also Supplementary Fig. 5). This might be attributed to modest effects of substrate binding on the conformation of the active site and/or the global protein conformation. The lack of any superhyperfine splitting from bromine nuclei argues against a direct cobalt–halogen bond formation in the [Co$^{II}$]-state. Upon reduction of PceA with 1 mM Ti(III) citrate and 20 μM methyl viologen, the [Co$^{II}$]-signal disappeared, indicating the formation of the EPR silent [Co$^{I}$]-state (Fig. 4, right panel). At the same time, a characteristic [4Fe–4S]$^{1+}$-signal appeared, which was combined with the spectra of excess Ti(III) citrate and methyl viologen radicals. Reduction of PceA mixed with excess substrate and incubation for about 5 min led to re-oxidation of the cobamide due to substrate conversion. Since the resulting spectra resembled those of the as-isolated enzyme, the oxidation state of the cobalt appeared to be similar. Neither the substrates nor the products caused a change in the [Co$^{II}$]-spectrum. Both observations seemed to exclude a direct coupling and thus strongly supported the hypothesis of an alternative mechanism, such as long-range electron transfer.

While none of the brominated phenols trapped in the crystal pointed with a bromine substituent towards the cobalt, PceA crystals soaked with TCE showed two substrate orientations, both exposing a chlorine substituent towards the cobalt[7]. A direct binding of the substrate-bound chloride was not observed, but it was unclear if either the substrate or leaving halide ion may move to bind the cobalt ion during catalysis. From molecular simulations for PceA–PCE/TCE complexes, Liao *et al.*[28] suggested the formation of a cobalt–halogen bond after heterolytic cleavage of the substrate–chloride bond, comparable to what was previously described for NpRdhA. However, addition of PCE and TCE in EPR analyses did not change the EPR signal of the as-isolated PceA (Fig. 4, magenta and cyan traces). In addition, the spectrum of a turnover-induced re-oxidized enzyme has largely reverted to the spectrum of the as-isolated PceA, contradicting a direct cobalt–halogen interaction. PCE is selectively reduced via TCE to *cis*-1,2-dichloroethene (*cis*-1,2-DCE). The reduced PceA sample incubated with *cis*-1,2-DCE showed a residual Ti(III) citrate signal and a methyl viologen radical signal consistent with the lack of measurable dehalogenation activity for this substrate (Fig. 4, black traces). Though *cis*-1,2-DCE is not converted by PceA, a low [Co$^{II}$]-amplitude is present in the EPR measurement after reduction and *cis*-1,2-DCE addition. This suggests that a single electron might be transferred from [Co$^{I}$] onto *cis*-1,2-DCE, resulting in the oxidation to [Co$^{II}$]. However, the energy required for the formation of vinyl chloride was calculated to be 29.3 kJ mol$^{-1}$ higher than for TCE reduction[28]. This energetic barrier is probably not broken by the *S. multivorans* PceA, so that the enzyme is not able to eliminate a chlorine substituent from *cis*-1,2-DCE. The energetic barriers for the formation of *trans*- and 1,1-DCE from TCE were calculated to be higher than that for *cis*-1,2-DCE formation, which may also prevent their formation[28]. An undirected formation of all isomers of DBE was described for the reduction of TBE, which are even further reduced to vinyl bromide by PceA[5] (Supplementary Fig. 1). The presence of bromine substituents rather than chlorine substituents might lower the energetic requirements as mentioned before. TBE and *cis*-/*trans*-DBE were tested in EPR measurements and led to the formation of the [Co$^{II}$]-state in the reduced sample, which is based on substrate conversion (Fig. 4,

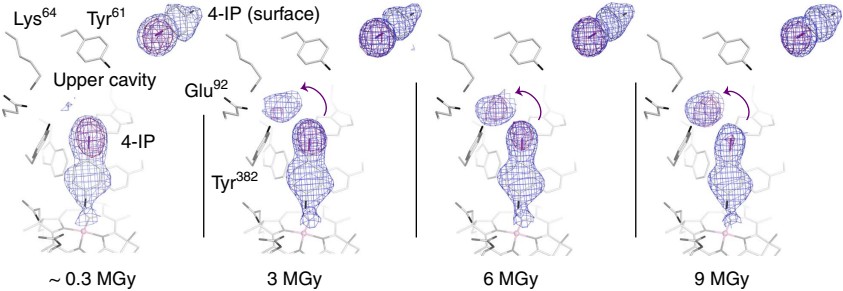

**Figure 5 | X-ray photon-induced dehalogenation.** Time-series of a crystal soaked in 4-iodophenol under non-reducing conditions, exposed to a 50 μm diameter synchrotron X-ray beam (approximately $10^{11}$ photons s$^{-1}$, $\lambda = 1.4$ Å) at 100 K. Both $2F_o$-$F_c$ (blue, 1 $\sigma$) and iodine-anomalous difference (purple, 5 $\sigma$) electron density, shown for the substituent, successively shifted away from C4 into the cavity around Lys64, Glu92 and Tyr382.

violet and gold traces). Again, no direct cobalt–halogen interaction was detected.

**Photon-induced dehalogenation in the PceA active site.** During the measurement of diffraction data, crystals were exposed to X-rays with approximate X-ray doses of $10^6$–$10^7$ Gy (Raddose-3D server[29]). Such radiation may induce photoreduction in metalloproteins (reviewed in ref. 30), whereby electrons radiolytically produced from the protein or solvent are trapped in the active site, potentially mimicking an electron transfer step[31]. Changes in the active site can precede global changes in the protein by orders of magnitude[32]. In crystals incubated with 4-IP, dissociation of the halide substituent was routinely visible in the electron density map (Fig. 5). When data were measured in a highly redundant manner, such that individual dose-dependent subsets of data could be used for structure solution, the iodine substituent clearly dissociated from 4-IP during data collection. Electron density of the substituent, weakened at the position of the leaving halide compared to the remainder of the substrate, and a nearby peak in the anomalous difference map, indicative of an appearing halide, were tracing a route for a leaving halide ion into the upper cavity and providing a model for the reductive halogen dissociation at C4. In contrast, a 4-IP bound to the surface of PceA did not show such photon-induced reduction (Fig. 5), which indicates that the active site architecture contributes to the observed dehalogenation reaction by trapping 4-IP in a state predestined to act as acceptor for free electrons within the protein. The source of such electrons differs and it is thus uncertain if the path taken by X-ray electrons follows that during catalysis. Although the possibility of X-ray-induced photoreduction of protein-bound cobamides from [$Co^{III}$] to [$Co^{II}$] has been reported before[33], the direct involvement of the super-reduced [$Co^I$] in the photon-induced dehalogenation of 4-IP remains elusive. A photon-induced dehalogenation of brominated or chlorinated phenols has not been observed as clearly as it was monitored for 4-IP.

**Discussion**
In the study presented here, the cobamide-containing and haloethene-converting PceA was shown to efficiently dehalogenate brominated phenols. Comparison of substrate–enzyme complexes allowed for conclusions on the initial attack by the cobamide cofactor. The distant positioning of the substrates with the hydroxyl group towards the cobalt and the absence of intimate cobalt–substrate interactions during conversion strongly suggested a long-range electron transfer mechanism in PceA. The super-reduced [$Co^I$] acts as reactive species, whereby electrons might be transferred to the phenol ring (Fig. 6a). Especially the halogen substituents in the *para*-position apparently must receive the electrons via the aromatic ring. The fate of the substrate is

then largely decided by internal charge distribution and the substitution pattern. According to the theory of substituent effects, the phenolic hydroxyl group shows a more positive resonance effect increasing the electron density particularly at the *ortho*- and *para*-positions, while the halogen substituents have a stronger inductive effect. The bromine substituent with the most positive $\sigma$ partial charge is predicted to be removed. The substrate 2,4,6-TBP is completely dehalogenated to phenol via 2,4-DBP and 4-BP. This observation is consistent with predictions derived from the natural bond orbital model based on the partial charges for each substituent[23]. The substrate radical formed upon the first electron transfer accompanied by the elimination of the halogen has to be neutralized by another electron transferred via [$Co^I$] or the proximal [4Fe–4S] cluster and a proton. The proton required to support substitution at the *para*-position C4 may be provided by a network of water molecules and ionizable side chains, including Tyr102 and Tyr382 in the upper cavity (Fig. 6b). A proton for neutralization at the *ortho*-positions C2 and C6 can be provided by Tyr246, pointing with its hydroxyl group towards the substrate at a distance of 4.3 Å from the bromine substituents or alternatively from solvent molecules in the substrate channel (Fig. 6c). In NpRdhA, mutation of the highly conserved Tyr246 equivalent inactivated the enzyme[8], which highlights its essential role. The leaving halide may move from C4 into the upper cavity and from C6 into the solvent channel. Movement of a bromide ion from C2 or C6 towards the cobalt or from C2 away from the substrate would require a concerted dissociation of the substrate from its position.

NpRdhA–substrate complex models using automatic docking or molecular dynamics simulations placed the substrate in orientations with either the bromine substituent *ortho* to the hydroxyl group above the cobalt or out of the axial position with the hydrogen in *meta*-position closest to the cobalt[8,34]. For the enzyme–substrate complex of NpRdhA with 3,5-dibromo-4-hydroxybenzoate, a direct interaction between substrate-bound bromine and cobalt was observed by EPR[8]. The 3,5-dibromo-4-hydroxybenzoate might be located closer to the cobalt in NpRdhA than 2,4,6-TBP in PceA, allowing a cobalt–halogen interaction in this enzyme. Recently, different mechanisms for the enzymatic reductive dehalogenation of chloroanilines were proposed for different organohalide-respiring bacteria[35]. Based on the variations in the range of products formed during substrate conversion combined with quantum chemical calculations, an initial attack on a halogen substituent or alternatively on a carbon-attached hydrogen atom was discussed. Hence, the mode of electron transfer from the cobalt to the substrate may vary between different RDases or even within one enzyme depending on the substrate. We show that in PceA phenolic substrates are excluded from direct interaction with [$Co^{II}$] and most likely [$Co^I$] by a tight packing of aromatic side chains that places the substrate at an angle of

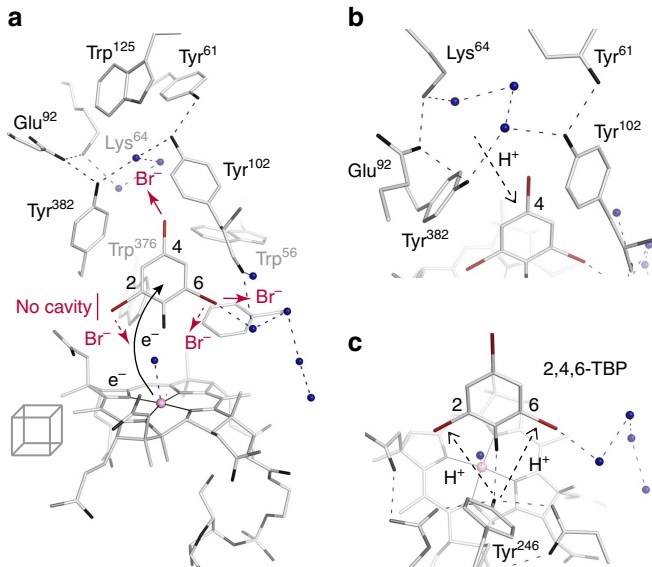

**Figure 6 | Potential electron and proton movements during halide ion release.** (**a**) Position of proximal iron–sulfur cluster, cobalt and 2,4,6-TBP substrate constrained by protein side chains. Within the constraints of the binding pocket, a leaving C2 substituent is not free to diffuse away from the substrate, a C4 halide may move to the upper cavity and a C6 halide may move to the substrate channel. It is not clear if the Co-water/hydroxyl ligand can be exchanged for a C2 or C6 halide with the substrate bound after reduction of PceA. (**b**) Exchangeable protons near C4 are present in the upper pocket in a network of ionizable side chains and water molecules. (**c**) Tyr246 may act as proton donor for C2 or C6. The view shows residues in front of panel **a**.

40° against the surface normal vector of the corrin ring, where it is kept at a distance of 4.7 Å from the cobalt. So far, only selected organohalides such as propyl iodide seem to overcome the structural barrier above the cobamide cofactor at the active site of PceA and directly interact with the cobalt. However, propyl iodide inhibited PceA function and slowed down substrate conversion as it has been shown for 4-BP or PCE[36]. Adduct formation in PceA was very recently proposed by Johannissen *et al.*[37], who reported on the probability of cobalt–halogen interaction between [Co[I]] and TCE based on density functional theory calculations. The instability of PceA crystals harbouring [Co[I]] and the substrate did not allow for approving this assumption for BPs and halogenated ethenes in the study presented here. However, the finding that none of the halogenated compounds showed a direct interaction between cobalt and substrate in EPR analyses rather points towards an alternative mechanism. This is furthermore supported by earlier reports about the formation of substrate radicals during the conversion of chlorinated propenes[4,16]. In addition, the appearance of [Co[II]] after incubation of super-reduced PceA with *cis*-1,2-DCE suggests that a single electron is transferred, although no vinyl chloride is formed.

The PceA structure does not seem to reduce the energetic barrier of *cis*-1,2-DCE dechlorination sufficiently. However, other TCE RDases have been described that convert *cis*-1,2-DCE further to vinyl chloride and ethene[38]. The molecular basis for these differences needs further investigation. Overall, PceA displays an unforeseen way of utilizing a cobamide cofactor for enzymatic reductive dehalogenation, which clearly differs from classical cobamide-dependent biochemistry. In order to evaluate the distribution and utilization of the three possible modes of initializing cobamide-mediated enzymatic reductive dehalogenation, more information from RDase structural and spectroscopic analyses is required.

## Methods

**Cultivation of bacteria.** *S. multivorans* (DSMZ 12446) and its mutant strain GD21 were cultivated anaerobically at 28 °C in a defined mineral medium[39] in the absence of exogenous cobamide and yeast extract (for generation of the mutant strain, see below). Pyruvate (40 mM) was used as an electron donor and PCE (10 mM nominal concentration) as an electron acceptor. PCE was added from a sterile and anoxic 0.5 M stock solution in hexadecane. The iron concentration in the medium was adjusted to 720 μM using an autoclaved and anoxic 144 mM $FeSO_4$ solution prepared in 50 mM $H_2SO_4$. For each experiment, two consecutive cultivations were performed, whereby the second culture served as inoculum (10%) for the main culture. Cells were harvested at an optical density of 0.26–0.28 (578 nm) by centrifugation (12,000*g*, 10 min, 10 °C) under aerobic conditions. The cell pellets were stored at − 20 °C.

**Purification of PceA–Strep.** The recombinant PceA–Strep was purified via affinity chromatography using Strep-Tactin Superflow column material (IBA, Göttingen, Germany). Since the membrane attachment of the enzyme is rather loose and PceA is sheared off during cell breakage, the enzyme was purified from the soluble and the membrane fraction. All steps were performed in an anaerobic glove box (CoyLab, Grass Lake, MI, USA). The cells of *S. multivorans* GD21 were resuspended in anoxic buffer A (100 mM Tris/HCl pH 8.0) (2 ml per 1 g wet cells) amended with protease inhibitor (cOmplete Mini EDTA-free Protease Inhibitor Cocktail; Roche Diagnostics, Mannheim, Germany). The cell lysate obtained after cell disruption in a French Press at 1,000 Psi (French Pressure Cell Press; Thermo Fisher Scientific, Germany) was subjected to ultracentrifugation (100,000*g*, 45 min, 4 °C; L8-M Ultracentrifuge, Rotor Ti70; Beckman Coulter, Krefeld, Germany). The supernatant, henceforth referred to as soluble extract, was transferred to a Strep-Tactin column (1 ml bed volume; IBA). PceA–Strep was eluted from the column using buffer A amended with 2.5 mM desthiobiotin (IBA). Tris(2-carboxyethyl)phosphine (TCEP) (Alfa Aesar, Karlsruhe, Germany) (5 mM) was added to protein samples applied to crystallization. Protein samples used for EPR spectroscopy contained 10% glycerol. PceA–Strep was concentrated via ultrafiltration in a Vivaspin 6 (30 K) centrifugal concentrator (Sartorius, Göttingen, Germany). The membrane pellet obtained from the ultracentrifugation (see above) was resuspended in buffer A (2 ml per 1 g pellet) and stirred overnight at 4 °C. Subsequently, the sample was again subjected to ultracentrifugation. The supernatant, henceforth referred to as membrane extract, was transferred to a Strep-Tactin column and PceA–Strep was purified and concentrated following the protocol described above. For crystallization, prePceA–Strep (bearing the twin-arginine translocation (Tat) signal peptide) and mature PceA–Strep (without the signal peptide) were separated using a Mono Q column (1 by 10 cm). The elution fractions were pooled and diluted 1:7 in basal buffer (50 mM Tris/HCl pH 7.5, 0.5 mM DTT amended with 5 mM TCEP) before application on the pre-equilibrated column. The mature enzyme eluted at 0.17 M NaCl in a stepwise gradient from 0 to 0.25 M in 5 column volumes (CV) and from 0.25 to 0.5 M NaCl in 2 CV. The elution buffer was replaced by the storage buffer (30 mM Tris-HCl, pH 7.5, 5 mM TCEP) by repeated concentration and resuspension of PceA. Structural analysis of PceA in complex with 3-BP was conducted with enzyme containing methoxybenzimidazolyl-norcobamide rather than norpseudo-B$_{12}$ as cofactor. Pure PceA enzyme was stored at − 80 °C. In total, 4 mg of purified PceA–Strep were obtained from 3 g cell protein. The protein concentrations were measured with the Bradford assay[40] using the Roti-Nanoquant reagent (Roth, Karlsruhe, Germany). For the separation of proteins, SDS–polyacrylamide gel electrophoresis (12.5%) was applied. The immunological analysis was conducted in accordance with John et al.[41]. The Strep-tag antibody solution (IBA) was diluted 3,000-fold and the antibodies were detected via a secondary antibody coupled to alkaline phosphatase (Sigma-Aldrich, Munich, Germany). A representative SDS–polyacrylamide gel electrophoresis with all purification fractions is shown in Supplementary Fig. 6. Further details about determination of the cofactor contents are given in Supplementary Table 2.

**Enzyme activity measurement.** All substrates used in this study were purchased from Sigma-Aldrich GmbH and abcr GmbH (Karlsruhe, Germany) in the highest purity available. Activity measurements of the PceA enzyme were conducted in HPLC vials (volume 1.5 ml) sealed with butyl rubber stoppers and flushed with nitrogen. The assay was performed in 100 mM Tris-HCl pH 7.5 at 22 °C. The artificial electron donor methyl viologen (0.5 mM) was reduced by adding 1.4 mM Ti(III) citrate. Pure PceA–Strep was present in concentrations of 3–36 nM for activity measurements with PCE and brominated phenols and 300–450 nM for chlorinated phenols. The organohalides were added from stock solutions in ethanol (80–100 mM) to get a final concentration of 0.5 mM in the assay mixture. After an incubation time of 1 min the reaction was stopped by rapid cooling of the reaction vessel to − 20 °C. To test for the influence of ammonium ions on the dehalogenating activity, 4 mM $(NH_4)_2SO_4$ were added. $K_m$ values were determined by fitting enzyme kinetic data to Michaelis-Menten kinetics. Activity assays with PceA crystals were performed in crystallization solution (100 mM Tris-HCl, pH 7.5, 200 mM sodium malonate, 2% (w/v) benzamidine, 25% (w/v) PEG 3350, 20% glycerol) amended with 0.5 mM methyl viologen and 1.4 mM Ti(III) citrate. Crystal fragments of a 1 mg ml[−1] PceA stock were added to the buffer. All enzyme activity measurements were performed in at least two biological replicates. Chlorinated ethenes were detected with a flame ionization detector coupled to a Clarus 500 Gas

Chromatograph (Perkin Elmer, Rodgau, Germany), which was equipped with a CP-PoraBOND Q FUSED SILICA 25 m × 0.32 mm column (Agilent Technologies, Böblingen, Germany). A headspace sample was taken from a 1 ml assay mixture incubated at 95 °C for 6 min. The chlorinated ethenes were separated under constant nitrogen flow in a temperature gradient from 4 min at 150 °C to 280 °C in 10 °C min$^{-1}$ steps (detector at 300 °C). Nonane was used as an internal standard. Retention times were as follows: PCE 9.8 min; TCE 6.9 min; cis-1,2-DCE 4.3 min. Halogenated phenols were separated using a reversed-phase HPLC system (Merck-Hitachi, Darmstadt, Germany) equipped with an RP8 column (LiChrospher 100, ID 4.6 × 100 nm, Merck, Darmstadt, Germany) and an ultraviolet/visible detector (210 nm). Also, 50% (v/v) methanol/0.3% (v/v) $H_3PO_4$ was used as a mobile phase with a flow rate of 0.4 ml min$^{-1}$. Retention times were as follows: 2,4,6-TBP 73.8 min; 2,4-DPB 34.9 min; 2,6-DPB 21.3 min; 2,5-DPB 31.8 min; 3,5-DPB 56.6 min; 2-BP 11.8 min; 3-BP 15.9 min; 4-BP 15.3 min; 4-IP 17.7 min; phenol 6.5 min; 2,3-DCP 22.2 min; 2,5-DCP 24 min; 3-CP 13.2 min; nd 4-CP 12.7 min. Non-enzymatic conversion of the substrates was tested with heat-inactivated PceA–Strep, which was incubated anoxically for 10 min at 95 °C before addition to the activity assay. The final concentration of heat-inactivated PceA–Strep was 526 nM, and the reaction mixture was incubated for 20 min. The inhibition of PceA–Strep (47 nM) converting 4-BP was tested by adding 100 μM propyl iodide before incubation of the assay mixture in the dark for 2 min.

**Crystallization and structure determination.** Crystallization, ligand incubation and flash cooling of crystals were performed under anoxic conditions in a glove box (model B; CoyLab) under an atmosphere of 95% $N_2$/5% $H_2$ and <10 p.p.m. oxygen. Crystals were grown by the sitting drop vapour diffusion method at room temperature. In all, 1 μl of 12 mg ml$^{-1}$ PceA in 30 mM Tris-HCl, pH 7.5, and 5 mM TCEP was mixed with 1 μl of crystallization solution containing 12–17% (w/v) PEG 3350 and 0.2 M sodium malonate, 2% benzamidine-HCl and 50 mM Tris-HCl, pH 7.5. Crystals were flash cooled in liquid nitrogen after protection in the crystallization solution supplemented with 20% (v/v) glycerol and 25% (final w/v) PEG 3350 (substrates 3-CP, 2,4-DBP, 2,6-DCP and 2,4,6-TCP). For all other substrates, a chloride-free cryo-incubation buffer was prepared from 50 mM Tris (free base), 2% benzamidine (free base), adjusted to pH 8 with malonic acid, followed by the addition of 25% PEG 3350 and 20% glycerol. Ten-fold concentrated substrate stock solutions were prepared in cryo-buffer at saturating concentrations and crystals were incubated with cryo-protectant/substrate-incubation buffer for 30–120 min. 4-BP and 2,4-DBP crystals were first reduced with 2 mM Ti(III)citrate and 0.2 mM methyl viologen. Once plunged into liquid $N_2$, crystals were removed from the anoxic atmosphere and from thereon stored and handled under liquid $N_2$.

Diffraction data were collected at 100 K on BL14.1 operated by the Helmholtz-Zentrum Berlin (HZB, Germany) at the BESSY II electron storage ring (Berlin-Adlershof, Germany)[42] at 1.9 Å X-ray wavelength for 3-CP, 4-CP, 2,6-DCP, 2,4,6-TCP and 2,4,5-TCP and at 0.91841 Å for all other substrates. Data were indexed and integrated with the XDS package[43] and XDSAPP[44]. Restraints for substrate ligands were prepared with eLBOW[45] (2,6-DCP, 2,4-DBP) and the Grade Server v1.001 (Global Phasing Ltd., Cambridge, UK). Models were fitted in COOT[46], refined with phenix.refine[47] and validated with Molprobity[48]. Nearly all of the residues (98%) were in the favoured region of the Ramachandran plot and no outliers were detected. Data collection and refinement statistics are summarized in Supplementary Table 3.

**Electron paramagnetic resonance spectroscopy.** EPR spectra were recorded on a Bruker ECS-106 X-band spectrometer equipped with home-built helium-cryogenics. Low-spin [Co$^{II}$] was typically detected at a temperature of circa 22–31 K and [4Fe–4S]$^{1+}$ clusters at circa 17 K. For substrate–cobalt interaction measurements, the enzyme (approximately 56 μM) was reduced by the addition of 1 mM Ti(III) citrate in combination with 20 μM methyl viologen as electron mediator. The substrates were added in a final concentration of 5 mM from stock solutions in ethanol. The maximal water solubility for 2,4,6-TBP, PCE and TBE is <1 mM, thus the EPR samples were oversaturated with substrate. All additions of substrates and/or reductant/oxidant were done anaerobically. Samples taken directly from the liquid nitrogen storage were prepared for additions by connecting the EPR tube to a vacuum/argon manifold with subsequent 10 vacuum/argon cycles until the onset of thawing and with all subsequent handlings under Argon 5.0 (0.2 bar overpressure). Additions of substrates and the reductant Ti(III) citrate were performed with Hamilton syringes from anaerobic solutions through the rubber connecting the EPR tube with the manifold. For the determination of the oxidation states of the cofactors of PceA (see Supplementary Fig. 7), EPR absorption spectra were recorded using PceA or PceA–Strep as isolated. The sample preparation was performed in an anaerobic chamber. For this purpose, 200 μl of the purified enzyme, stored in Tris-HCl buffer (50 mM Tris-HCl (pH 7.5), 10% (v/v) glycerol) were transferred to an EPR tube and frozen in liquid nitrogen. In parallel, the species of the iron–sulfur clusters was determined in 200 μl samples of purified enzyme reduced by the addition of 10 mM sodium dithionite. Sodium dithionite was added as 200 mM solutions in anoxic Tris-HCl buffer (50 mM Tris-HCl (pH 7.5), 10% glycerol). For oxidation of the sample, 0.5 mM $K_3$[Fe(CN)$_6$] was added to the 200 μl sample. After incubation for 5 min, the samples were frozen in liquid nitrogen and stored at −80 °C until measurement. EPR spectra were recorded with modulation amplitude of 8 Gauss and a modulation frequency of 100 kHz. Spin quantification was done versus an external copper standard (10 mM CuSO$_4$, 10 mM HCl, 2 M NaClO$_4$) as described previously[49].

**Plasmid construction.** All enzymes used for DNA modification in this study were purchased from Fermentas (St Leon-Rot, Germany) or New England Biolabs (Ipswich MA, USA). All plasmids used in this study are summarized in Supplementary Table 4. The cloning steps were conducted according to standard techniques described in Sambrook et al.[50] using the Escherichia coli strain Dh5α. The plasmid pY179 (ref. 51) was used as the starting material. This plasmid is a derivative of pBluescript II SK+ (Stratagene, La Jolla CA, US). Plasmids were extracted using the GeneJET Plasmid Miniprep Kit (Thermo Scientific, Darmstadt, Germany). The plasmid pY179 contains a 6-kb EcoRI DNA-fragment derived from S. multivorans genomic DNA. The DNA-fragment encloses the pceAB gene cluster. In order to reduce the size of the subcloned DNA fragment (3.2 kb) to almost only the pceAB gene cluster, the plasmid pY179 was cut with the restriction enzymes XhoI and BglII, treated with Klenow fragment and re-ligated. The resulting construct of this initial modification, plasmid pTOS024, was provided as template in an inverse PCR reaction conducted with the following primer pair T68/T69: 5′-TATGGCTAGCCATCACCATCACCATCACTCATGAAATTATTAAATATTT-TAAATTATAAAGCG-3′ and 5′-ATCGGCTAGCTGATTTTTTAACCCTA-TCCTTTC-3′. Using this method, a DNA sequence encoding a 6 × His-tag was fused to the 3′-end of the DNA sequence encoding the pceA gene. As a result of the inverse PCR reaction, a NheI-restriction site was generated at both ends of the PCR product. After the restriction with NheI, the PCR fragment was circularized and plasmid pTOS036 was formed. Subsequently, the DNA sequence encoding the 6 × His-tag was replaced by a DNA sequence encoding a Strep-tag II. For this purpose, a 250-bp PCR fragment was generated using plasmid pY179 as template and the oligonucleotides T110/AN38: 5′-TATGGCTAGCTGGAGCCAC-CCGCAGTTCGAAAAATCATGAAATTATTAAATATTTTAAATTA-TAAAGCG-3′ and 5′-GCGATCTAGCTCAAAAGAGAG-3′. The PCR fragment was cut with NheI and AflII and ligated into the similarly cut pTOS036 yielding plasmid pTOS071.

For the transformation of S. multivorans, the plasmid pBR322 (ref. 52) was used as carrier. In order to generate a homologous DNA sequence within this plasmid, which would allow for recombination into the pceAB gene cluster in the S. multivorans genome, a 3.2-kb DNA-fragment was cut from plasmid pY179 with BglII and BamHI and ligated into pBR322 cut with BamHI, yielding plasmid pTOS001. The subcloned DNA fragment contained the complete pceAB gene cluster with the upstream and downstream intergenic regions, including the promoter and terminator sequences of pceAB. As a selective marker for proving successful homologous recombination, the kanamycin-resistance cassette from plasmid pUC4K[53] was used. For this purpose, plasmid pUC4K was cut with BamHI. The resulting 1.3-kb DNA fragment was treated with Klenow fragment and ligated into plasmid pTOS001 cut with BstXI and treated with Klenow fragment. In the resulting plasmid pTOS012, the kan$^R$ cassette was located in the intergenic region downstream of the pceAB gene cluster and orientated against the pceAB genes. In order to transfer the modified DNA-sequence encoding the C-terminal Strep-tagged PceA from plasmid pTOS071 into plasmid pTOS012, pTOS071 was cut with PmlI and AflII and the resulting 620-bp DNA fragment ligated into the similarly cut pTOS012. Finally, plasmid pTOS077 was generated.

**Transformation of S. multivorans.** The protocol for transformation of S. multivorans was adapted from the procedure described by Simon et al.[54] for the transformation of W. succinogenes. Cells of S. multivorans were cultivated in anoxic medium containing 40 mM pyruvate as an electron donor and 40 mM fumarate as an electron acceptor. The cells were harvested in the exponential growth phase and washed twice with a sucrose (0.5 M)/glycerol (10% v/v) solution; finally the cells were resuspended in this solution (approximately 10 g protein l$^{-1}$). The electroporation of the cells and all subsequent steps were conducted in an anaerobic chamber (CoyLab). An aliquot of S. multivorans cells (40 μl) was mixed with plasmid DNA (approximately 1 μg) and transferred into an electroporation cuvette (0.2 cm pathlength; Bio-Rad, Hercules, CA, USA). The plasmid DNA was purified by precipitation with ammonium acetate[55]. The electroporation was performed in a Gene Pulser XCell (Bio-Rad, Hercules, CA, USA) under the following conditions: 25 μF, 2.5 kV, 200 Ω. Immediately after application of the pulse, 1 ml anoxic pyruvate/fumarate-containing medium was added to the cell suspension. After an incubation period of at least 1 h at 28 °C, the cells were transferred to solid medium. The solid medium contained all the ingredients of the liquid medium described above plus 1% (w/v) washed agar (AppliChem, Darmstadt, Germany) and 0.2% yeast extract. It was amended with 100 μg ml$^{-1}$ kanamycin. The plated cells were incubated for 1–2 weeks at 28 °C. Using plasmid pTOS077 for the electroporation, the mutant strain S. multivorans GD21 (PceA–Strep) was generated (Supplementary Fig. 8). The transformation efficiency was very low (a single transformant per 10 μg plasmid). The application of linear DNA in the transformation procedure did not have a positive effect on the transformation efficiency.

**Southern hybridization.** Genomic DNA of *S. multivorans* was isolated by phenol–chloroform isoamylalcohol extraction[56]. The DNA (30 µg) was cut with *Eco*RI, *Eco*RV and *Xmn*I. After inactivation of the restriction enzymes (65 °C, 20 min), the fragmented genomic DNA was loaded onto a 0.8% agarose gel prepared with 0.5-fold Tris-borate-EDTA buffer (44.5 mM Tris, 44.5 mM boric acid, 1 mM EDTA). For subsequent evaluation of the gel, a Digoxigenin-labelled DNA molecular weight marker (Roche Diagnostics) was applied. The gel was run for 3 h at 90 V. The separated DNA fragments were blotted onto a nylon membrane (Boehringer, Mannheim, Germany) using a semi-dry blot device (Bio-Rad Laboratories, Munich, Germany). The current applied was $3\,mA\,cm^{-2}$ for 1 h. Subsequently, the membrane was washed for 5 min in twofold saline–sodium citrate (SSC) buffer (30 mM Na–citrate pH 7.0, 0.3 M NaCl). The membrane was transferred into the denaturing solution (0.4 M NaOH, 10 mM EDTA, 1.5 M NaCl). After 10 min incubation at room temperature, the membrane was rinsed with $2 \times$ SSC and dried. The DNA fragments were crosslinked with the membrane by ultraviolet-light exposure. The membrane was incubated for 2 h at 60 °C in hybridization solution ($5 \times$ SSC containing 0.1% lauroylsarcosine and 0.02% SDS). The *pceA* and the *kan* DNA probes were diluted 1,000-fold in hybridization solution before their application to the membrane. The DNA probes were generated as follows: The primers 5′-AACCTTGGTTTTTATCAGCATATG-3′ and 5′-GGTCTTCTATCTAACCCTACTG-3′ were used to amplify the DNA fragment for the *pceA* probe and the oligonucleotides 5′-TGAGCCATATTC-AACGGG-3′ and 5′-GAATGCTGTTTTCCCGGG-3′ for synthesis of the *kan* probe via PCR. The DNA probes were labelled with Digoxigenin using the DIG DNA Labelling Kit (Roche, Mannheim, Germany) according to the manufacturer's instructions. The hybridization of the blotted DNA fragments with the DNA probes was conducted in a hybridization oven (HB-1,000 Hybridizer, UVP-Laboratory Products, Cambridge, UK) overnight at 65 °C. Subsequently, the membrane was washed twice for 5 min at room temperature (RT) in $2 \times$ SSC, 0.1% SDS, followed by two washing steps for 15 min at RT and 42 °C, respectively, in $0.5 \times$ SSC, 0.1% SDS. After incubation for 1 h in 1% blocking reagent (Roche Diagnostics), the membrane was washed three times for 10 min in MST-buffer (0.1 M maleic acid pH 7.5, 150 mM NaCl, 0.05% Tween 20). Subsequently, the membrane was incubated for 2 h in MST-buffer containing Anti-Digoxigenin-Alkaline Phosphatase, Fab-Fragments (Roche Diagnostics) that were diluted 5,000-fold. The membrane was washed twice with MST-buffer for 10 min. After a washing step in the developing buffer (0.1 M Tris/HCl pH 9.5, 0.1 M NaCl, 50 mM $MgCl_2$), the blot was stained in developing buffer containing 0.34 mg nitroblue tetrazolium and 0.175 mg 5-bromo-4-chloro-3-indolyl phosphate (Roth) per ml solution.

**Data availability.** Model coordinates and structure factors for ligand-bound PceA have been submitted to the protein data bank (PDB) under accessions numbers 5M2G (2,4,6-TBP), 5M8U (4-BP), 5M8W (4-CP), 5M8X (2,4,5-TCP), 5M8Y (3-CP), 5M8Z (2,3-DFP), 5M90 (3,4,5-TFP), 5M91 (2,6-DBP), 5M92 (2,4-DBP), 5MA2 (4-IP), 5MAA (3-BP), 5MA0 (2,6-DCP) and 5MA1 (2,4,6-TCP).

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

## Acknowledgements

This work was financially supported by the DFG Research Unit FOR1530 (grant DI 314/14-2), the Jena School for Microbial Communication (JSMC) and the Ernst Abbe Foundation. Peggy Brand-Schön is acknowledged for excellent technical assistance. Martin Bommer was financed by the DFG Collaborative Research Centre SFB 1078. We acknowledge access to beamlines of the BESSY II storage ring (Berlin, Germany) via the Joint Berlin MX-Laboratory sponsored by the Helmholtz Zentrum Berlin für Materialien und Energie, the Freie Universität Berlin, the Humboldt-Universität zu Berlin, the Max-Delbrück-Centrum and the Leibniz-Institut für Molekulare Pharmakologie.

## Author contributions

G.D. and T.S. conceived and coordinated the study. C.K. performed purification, biochemical characterization and activity assays with PceA. M.B. performed in-crystal substrate-binding experiments and built the ligand-bound structures. W.R.H. performed metal-cofactor analysis and substrate–cobalt interaction studies using EPR. T.S., C.K. and M.U. were involved in the generation and characterization of the *S. multivorans* mutant strain GD21. All authors participated in data analysis, discussion and writing of the manuscript.

## Additional information

**Competing interests:** The authors declare no competing financial interests.

