## [Peer Review File · Nature Communications]

Reviewers' Comments:

Reviewer #1 (Remarks to the Author):

This is an intriguing paper that suggests that the reductive dehalogenase PceA from *Sulfurospirillum multivorans* may use long-range electron transfer to remove halide ions from aryl halides. The proposed mechanism contrasts greatly from that previously suggested for a related dehalogenase, NpRdhA, from *Nitratireductor pacificus*. In general, I do think the data provided support the conclusions the authors have made, thus suggesting a "one-mechanism-fits-all" approach to this group of enzymes may not be appropriate. Vitamin B12 chemistry was thought for many years to be rather restricted in its roles/abilities, so it is definitely interesting to read about new developments in the field that expand the role of this cofactor in biochemistry. I think/hope that this work could be relatively easily repeated with the right expertise. The authors provide a great deal of helpful, useful methodology and do a fine job overall of explaining their research.

I do have a few questions/concerns that I think could improve the paper somewhat. These are listed below:

- 1) It really would have helped me, personally, and I think it would help other readers to include a Figure illustrating some of the proposed reactions, particularly since it seems like PceA accepts a much wider range of substrates than NpRdhA.
- 2) Similarly, somewhere in the introductory material, perhaps on p. 3, it would have been helpful to know what the sequence identity/similarity is between PceA and NpRdhA. (I know the authors already make some mention of similarity in the active sites.)
- 3) On page 3, the paragraph spanning lines 69-75 should probably be checked for additional references. There is certainly growing evidence that not all cobalamin-dependent enzymes (namely the ones that also depend on radical S-adenosyl-L-methionine chemistry) follow such a general paradigm.
- 3a) In line 135, why is the abbreviation for tetrachloroethylene apparently listed as "(PCE)"?
- 4) Line 138, "preferably" should probably say "preferentially."

- 5) Lines 150-153 should probably be reworded and/or combined because they are somewhat redundant. I think in line 150, the authors should say "has not been previously reported" or something similar.
- 6) I suggest improving the wording of lines 155-162 because the grammar and thus the meaning/interpretation are not coming through correctly. In particular, "lying beyond the conversion of the PCE" (line 161) does not quite make sense.
- 7) I suggest changing line 184 to say, "Propyl iodide is known inhibitor that binds irreversibly..."
- 8) I think lines 193-196 should be rewritten/reworded. What is a "molecular size?"
- 9) Line 199 - I suggest "analyzed" be changed to "determined and analyzed."
- 10) Line 201 "approx." should not be abbreviated.
- 11) Lines 226 and 227 should be reworded. As currently written, I can't tell whether this statement means that the Km supports the crystallography results and/or contradicts the prior sentence regarding bulk/steric hindrance?
- 12) Line 245: I don't think the word "challenge" is appropriate because "challenge" usually means disprove. Why would the authors want to disprove their own results?
- 13) Line 248: "are" should be changed to "were."
- 14) Line 267: "preferably" should be changed to "preferentially."
- 15) Line 280: "on" should be "of." I also suggest rewording the sentence that begins on line 280 to say something about "influences" or "determines" instead of "decide on the removal"...
- 16) I suggest rewording/reorganizing lines 283-287 to improve grammar and the reader's understanding.
- 17) Line 307: Please change "assumption" to "hypothesis" or a synonym.
- 18) Please correct the wording of lines 310-312 regarding the "overlaid spectra" - the spectra INCLUDED those signals as opposed to having other spectra laid on top of them, correct?
- 19) I don't think the word "supporting" in line 316 is correct. I think this sentence should be modified to reflect the fact that SINCE the spectra resembled those of the as-isolated enzyme, the

oxidation state of cobalt appeared to be similar, which thus seemed to rule out direct electron transfer and suggested an alternative mechanism such as long-range transfer.

20) Should lines 326-329 be accompanied by a figure reference? Line 330 should include the word "signal" after "citrate." Line 334 "suggest," should be "suggests" and the comma should be deleted.

21) Lines 344-347 also seem like they are missing a figure reference.

22) I suggest changing "have to" to "apparently must" or something similar in line 382. I also feel that the sentences in lines 383-387 are a bit redundant compared with the "results." Perhaps the "results" in general should be made more streamlines so that the "results" and "discussion" have less overlap.

23) Lines 402-403, "exchange...by Phe" should be changed to "mutation" or related wording.

24) Line 408 and subsequently elsewhere (such as line 414), "hydroxylbenzoate" should be "hydroxybenzoate."

25) I suggest improving/changing the wording for lines 424-427 because those sentences do not quite make sense.

26) Line 438: "and only energetic" doesn't seem to be the right wording.

27) Line 442: "investigations" should be singular "investigation" instead.

28) Lines 463 and 464 of the methods seem to be out of place. Should these go earlier in the cultivation methods paragraph?

29) Line 468: Please check references to "CoyLab" and "COY Labs" later in the document for consistency.

30) Line 471: Is "cOmplete" a typo?

31) Lines 476-482 seem like too much unnecessary information.

32) I did not quite understand why it seems like the same enzyme is purified from both the soluble and membrane fractions. Is there a difference between the two enzymes? There is definitely more information provided in the supplementary info that I think could partially be moved to this section to improve clarity. Also, the yield per liter culture or per gram of cell paste

should be reported here instead of in the supplementary information.

33) Line 526: "at least in two" should be corrected for grammar.

34) Line 533: "For testing an abiotic conversion" should be reworded. Is this a nonenzymatic control? Why heat inactivate the enzyme?

35) The materials/sources listed at the end of this section (lines 538-540) should be moved earlier in the paragraph.

36) Line 566: Delete "the" in front of "supplementary"

37) Line 577: What does "in excess" mean? Can a range of concentrations be listed?

38) Please clarify line 581, "under 0.2 bar overpressure of Ar 5.0" - what does this mean?

39) Are the "et al." designations in the references acceptable for this journal? Also, please check subscripts/lack of subscripts for "B12" throughout the references.

40) Table 1 - to what does "nkat" refer? Also, line 739, "kcat" should be lowercase.

41) Line 775, suggest changing the word "delimited"

42) Line 787, suggest adding "versus" or "compared with" between the angstrom resolutions.

43) Line 808, again suggest "overlaid" be changed to "accompanied by" or "additional signals from citrate/methyl viologen appeared"

Reviewer #2 (Remarks to the Author):

The paper by Kunze and colleagues follows up previous papers that first reported the X-ray crystal structures of cobamide-containing reductive dehalogenases (refs. 7 and 8). Specifically, the current paper reports detailed enzymological and biophysical studies using EPR spectroscopy and X-ray crystallography to study the dehalogenation reaction. Strikingly, the cobalt ion of the cofactor is involved in electron transfer chemistry required for dehalogenation, and the sum total of evidence presented in this paper strongly suggests that the substrate does not coordinate to the

metal ion during the course of the reaction, thus supporting a mechanistic model invoking through-space electron transfer between substrate and metal ion. As the authors note in the final sentence of the Abstract, such a mechanism "is without parallel in vitamin B12-dependent biochemistry".

The information in Fig. 4 is particularly intriguing, since it shows the time-dependent dehalogenation of a substrate. While it is generally appreciated that exposure to X-rays can trigger the photodissociation of carbon-halogen bonds in crystals, this reviewer is unaware of any crystallographic studies showing such beautiful "snapshots" of such a reaction.

This manuscript is suitable for publication in Nat. Commun. with only minor revisions:

1. On page 9, line 214, the authors refer to a "T shaped pi stack"; when aromatic rings interact in edge-to-face manner, they are really not "stacked".

2. Supplementary Table 4: the authors should consider the number of significant figures in the data reported. For example, it is not too meaningful to report thermal B-factors and Wilson B factors to the hundredths place; rounding to the "ones column" would be appropriate. R-merge and R-meas are reported with 5 significant figures overall, but only 4 significant figures for the outer resolution shell; perhaps only 3 significant figures would be sufficient here? Finally, the authors define the resolution cutoff as $\text{mean } I/\sigma > 2.0$ at the bottom of the table, yet the dataset 5MA1 has $I/\sigma = 1.59$. With $CC1/2 = 0.55$, the resolution cutoff of 2.5 Å for this structure determination might be a little optimistic. I also suggest being consistent with significant figures in listing resolution ranges. For example, dataset 5M2G has a resolution range of 47.26 - 1.8 Å, whereas dataset 5MA1 has resolution range 33.94 - 2.498 Å. Was the outer resolution limit not as precisely determined for 5M2G, or was the outer resolution limit arbitrarily rounded to 1.8 Å? At any rate, it is unusual for resolutions to be reported to the thousandths place, so it would be sufficient to consistently round these numbers to the hundredths place.

Response to Reviewers' Comments:

We are grateful for the review of our manuscript and the advice given.

Reviewer #1 (Remarks to the Author):

1) It really would have helped me, personally, and I think it would help other readers to include a Figure illustrating some of the proposed reactions, particularly since it seems like PceA accepts a much wider range of substrates than NpRdhA.

In order to improve the understandability, the proposed mechanisms of the initial attack of [Co^I] onto the halogenated substrate are depicted in a scheme now included in the Introduction of the revised text. Furthermore, the diversity of PceA substrates and their dehalogenation products is visualized in a new figure added to the Supplementary Information.

2) Similarly, somewhere in the introductory material, perhaps on p. 3, it would have been helpful to know what the sequence identity/similarity is between PceA and NpRdhA. (I know the authors already make some mention of similarity in the active sites.)

The amino acid sequence identity of 28% is mentioned in the introduction of the revised text.

3) On page 3, the paragraph spanning lines 69-75 should probably be checked for additional references. There is certainly growing evidence that not all cobalamin-dependent enzymes (namely the ones that also depend on radical S-adenosyl-L-methionine chemistry) follow such a general paradigm.

Additional references were added to the revised text and the description of the diversity of cobalamin-dependent enzymes was extended by including recent publications about cobalamin-dependent S-adenosylmethionine radical enzymes.

3a) In line 135, why is the abbreviation for tetrachloroethylene apparently listed as "(PCE)"?

The abbreviation PCE is derived from perchloroethylene, which is synonymously used for tetrachloroethene. Perchloroethylene is mentioned in the revised text, when the abbreviation PCE was used the first time.

4) *Line 138, "preferably" should probably say "preferentially."*

The text was modified as suggested by the reviewer.

5) *Lines 150-153 should probably be reworded and/or combined because they are somewhat redundant. I think in line 150, the authors should say "has not been previously reported" or something similar.*

Part of the statement was deleted to reduce redundancy and the text was modified as suggested by the reviewer.

6) *I suggest improving the wording of lines 155-162 because the grammar and thus the meaning/interpretation are not coming through correctly. In particular, "lying beyond the conversion of the PCE" (line 161) does not quite make sense.*

The respective paragraph was reworded to improve the understandability.

7) *I suggest changing line 184 to say, "Propyl iodide is known inhibitor that binds irreversibly..."*

The text was modified as suggested by the reviewer.

8) *I think lines 193-196 should be rewritten/reworded. What is a "molecular size?"*

The respective sentence was reworded and "molecular size" was replaced by "dimensions".

9) *Line 199 - I suggest "analyzed" be changed to "determined and analyzed."*

"Determined" was added to the revised text.

10) *Line 201 "approx." should not be abbreviated.*

The text was modified as suggested by the reviewer.

11) *Lines 226 and 227 should be reworded. As currently written, I can't tell whether this statement means that the Km supports the crystallography results and/or contradicts the prior sentence regarding bulk/steric hindrance?*

The sentence was reworded in order to improve clarity.

12) *Line 245: I don't think the word "challenge" is appropriate because "challenge" usually means disprove. Why would the authors want to disprove their own results?*

“Challenge” was replaced by “verify”.

13) *Line 248: "are" should be changed to "were."*

The text was modified as suggested by the reviewer.

14) *Line 267: "preferably" should be changed to "preferentially."*

“Preferably” was replaced by “preferentially”.

15) *Line 280: "on" should be "of." I also suggest rewording the sentence that begins on line 280 to say something about "influences" or "determines" instead of "decide on the removal"...*

The text was modified as suggested by the reviewer.

16) *I suggest rewording/reorganizing lines 283-287 to improve grammar and the reader's understanding.*

The paragraph was reworded.

17) *Line 307: Please change "assumption" to "hypothesis" or a synonym.*

We now use the word "hypothesis".

18) *Please correct the wording of lines 310-312 regarding the "overlaid spectra" - the spectra INCLUDED those signals as opposed to having other spectra laid on top of them, correct?*

The authors agree with the reviewer and the wording was corrected.

19) *I don't think the word "supporting" in line 316 is correct. I think this sentence should be modified to reflect the fact that SINCE the spectra resembled those of the as-isolated enzyme, the oxidation state of cobalt appeared to be similar, which thus seemed to rule out direct electron transfer and suggested an alternative mechanism such as long-range transfer.*

The sentence was modified as suggested by the reviewer.

20) *Should lines 326-329 be accompanied by a figure reference? Line 330 should include the word "signal" after "citrate." Line 334 "suggest," should be "suggests" and the comma should be deleted.*

The text was modified as suggested by the reviewer and a figure reference was added.

21) *Lines 344-347 also seem like they are missing a figure reference.*

A figure reference was included.

22) *I suggest changing "have to" to "apparently must" or something similar in line 382. I also feel that the sentences in lines 383-387 are a bit redundant compared with the "results." Perhaps the "results" in general should be made more streamlines so that the "results" and "discussion" have less overlap.*

This part was shortened to avoid overlap with the results.

23) *Lines 402-403, "exchange...by Phe" should be changed to "mutation" or related wording.*

The sentence was modified as suggested by the reviewer.

24) *Line 408 and subsequently elsewhere (such as line 414), "hydroxylbenzoate" should be "hydroxybenzoate."*

The spelling mistakes were corrected in the revised manuscript.

25) *I suggest improving/changing the wording for lines 424-427 because those sentences do not quite make sense.*

Part of the sentences was removed to improve clarity.

26) *Line 438: "and only energetic" doesn't seem to be the right wording.*

The sentence was reworded.

27) *Line 442: "investigations" should be singular "investigation" instead.*

The "s" was removed.

28) *Lines 463 and 464 of the methods seem to be out of place. Should these go earlier in the cultivation methods paragraph?*

The sentence was removed, since cultivation experiments with bromophenols are not part of the revised manuscript.

29) *Line 468: Please check references to "CoyLab" and "COY Labs" later in the document for consistency.*

We now used consistently CoyLab.

30) *Line 471: Is "cOplete" a typo?*

It is not. It is the commercial name of the protease inhibitor cocktail mix purchased by Roche Diagnostics (Mannheim, Germany).

31) *Lines 476-482 seem like too much unnecessary information.*

The information was removed from the revised manuscript.

32) *I did not quite understand why it seems like the same enzyme is purified from both the soluble and membrane fractions. Is there a difference between the two enzymes? There is definitely more information provided in the supplementary info that I think could partially be moved to this section to improve clarity. Also, the yield per liter culture or per gram of cell paste should be reported here instead of in the supplementary information.*

Since the membrane attachment of the enzyme is rather loose and PceA is sheared off during cell breakage, the enzyme was purified from the soluble and the membrane fraction. 4 mg of purified PceA-Strep were obtained from 3 g cell protein. The information is now part of the revised text.

33) *Line 526: "at least in two" should be corrected for grammar.*

Was changed to "...were performed in at least two biological replicates."

34) *Line 533: "For testing an abiotic conversion" should be reworded. Is this a nonenzymatic control? Why heat inactivate the enzyme?*

This is a non-enzymatic control. Several organohalides can also be dehalogenated abiotically by the cobamide-cofactor alone. The cobamide cofactor is heat resistant and remains in the boiled protein sample. Thus, the use of heat-inactivated protein is a control to test for the proportion of abiotic conversion. The main text was changed accordingly.

35) *The materials/sources listed at the end of this section (lines 538-540) should be moved earlier in the paragraph.*

The sources of all materials is now mentioned at the beginning of this paragraph.

36) *Line 566: Delete "the" in front of "supplementary"*

Done.

37) *Line 577: What does "in excess" mean? Can a range of concentrations be listed?*

The substrates were added in a concentration of 5 mM from a stock solution in ethanol. However, water solubility for most substrates is lower (≤ 1 mM), thus oversaturated concentrations were present in the sample. This information is now part of the revised text.

38) *Please clarify line 581, "under 0.2 bar overpressure of Ar 5.0" - what does this mean?*

The phrase was reorganized to improve clarity and the abbreviation Ar (argon) is now explained in the revised text.

39) *Are the "et al." designations in the references acceptable for this journal? Also, please check subscripts/lack of subscripts for "B12" throughout the references.*

Nature Communication style: If there are six and more authors, only the first author should be mentioned followed by *et al.* The references were checked for consistency.

40) *Table 1 - to what does "nkat" refer? Also, line 739, "kcat" should be lowercase.*

"nkat mg⁻¹" refers to PceA, which is now stated in the revised text. The text was modified as suggested by the reviewer.

41) *Line 775, suggest changing the word "delimited"*

"Delimited" was replaced by "encircle".

42) *Line 787, suggest adding "versus" or "compared with" between the angstrom resolutions.*

"Versus" was added between the resolutions.

43) Line 808, again suggest "overlaid" be changed to "accompanied by" or "additional signals from citrate/methyl viologen appeared"

"Overlaid" was changed to "accompanied by".

Reviewer #2 (Remarks to the Author):

1. On page 9, line 214, the authors refer to a "T shaped pi stack"; when aromatic rings interact in edge-to-face manner, they are really not "stacked".

We have rephrased the perpendicular interaction of two aromatic rings in a more descriptive way: "The hydrogen at C3 points at the Tyr382 phenyl group. The two rings interact in an edge-to-face geometry, precluding a bulky halogen substituent at this place."

2. Supplementary Table 4: the authors should consider the number of significant figures in the data reported. For example, it is not too meaningful to report thermal B-factors and Wilson B factors to the hundredths place; rounding to the "ones column" would be appropriate. R-merge and R-meas are reported with 5 significant figures overall, but only 4 significant figures for the outer resolution shell; perhaps only 3 significant figures would be sufficient here? Finally, the authors define the resolution cutoff as $\text{mean } I/\sigma > 2.0$ at the bottom of the table, yet the dataset 5MA1 has $I/\sigma = 1.59$. With $CC1/2 = 0.55$, the resolution cutoff of 2.5 Å; for this structure determination might be a little optimistic. I also suggest being consistent with significant figures in listing resolution ranges. For example, dataset 5M2G has a resolution range of 47.26 - 1.8 Å; whereas dataset 5MA1 has resolution range 33.94 - 2.498 Å;. Was the outer resolution limit not as precisely determined for 5M2G, or was the outer resolution limit arbitrarily rounded to 1.8 Å;? At any rate, it is unusual for resolutions to be reported to the thousandths place, so it would be sufficient to consistently round these numbers to the hundredths place.

We have updated the crystallographic data collection and refinement Table according to the suggestions made by the reviewer and the journal guidelines. The statistics for 5MA1 have been recalculated the reported resolution is now reported as 2.6 Å instead of 2.5 with $I/\sigma(I)$ of 2.3 for the highest resolution shell.

(Just to answer the question, disparities in the number of significant figures for resolution of different structure came from the use of an automated pipeline, XDSAPP; and then where this did not give a satisfactory cut-off, manual XDS processing was used with a cut-off to one decimal point only.)